# Comparative Study on the Densification, Microstructure and Properties of WC-10(Ni, Ni/Co) Cemented Carbides Using Electroless Plated and Coprecipitated Powders

**DOI:** 10.3390/ma16051977

**Published:** 2023-02-28

**Authors:** Haoli Jiang, Jing Tong, Zhaoqing Zhan, Zhanhu Yao, Songbai Yu, Fanlu Min, Congxu Wang, Jacques Guillaume Noudem, Jianfeng Zhang

**Affiliations:** 1College of Mechanics and Materials, Hohai University, Nanjing 211100, China; 2CCCC Tunnel Engineering Company Limited, Beijing 100102, China; 3Key Laboratory of Geomechanics and Embankment Engineering, Hohai University, Nanjing 210098, China; 4Shaanxi Aeronautic Carbide Tool Co., Ltd., Hanzhong 724200, China; 5CRISMAT-ENSICAEN (UMR-CNRS 6508), Université de Caen-Basse-Normandie, F-14050 Caen, France

**Keywords:** cemented carbide, chemical coating method, WC-Ni/Co, mechanical properties, corrosion resistance

## Abstract

More and more attention is being paid to the influence of powder mixing on the mechanical properties and corrosion resistance of WC-based cemented carbides. In this study, WC was mixed with Ni and Ni/Co, respectively, by chemical plating and co-precipitated-hydrogen reduction, which are labelled as WC-Ni^EP^, WC-Ni/Co^EP^, WC-Ni^CP^ and WC-Ni/Co^CP^, respectively. After being densified in a vacuum, the density and grain size of CP were denser and finer than those of EP were. Simultaneously, the better mechanical properties of flexural strength (1110 MPa) and impact toughness (33 kJ/m^2^) were obtained by WC-Ni/Co^CP^ due to the uniform distribution of WC and binding phase and solid solution enhancement of the Ni-Co alloy. In addition, the lowest self-corrosion current density of 8.17 × 10^−7^ A·cm^−2^, a self-corrosion potential of −0.25 V and the biggest corrosion resistance of 1.26 × 10^5^ Ω in 3.5 wt % NaCl solution were obtained by WC-Ni^EP^ because of the presence of the Ni-Co-P alloy.

## 1. Introduction

Due to the comprehensive properties of hardness, strength and toughness, tungsten carbide (WC)-based cemented carbides have been realized as one of the most widely used powder metallurgy products to meet various engineering requirements in the world, such as those of marine construction [1,2,3,4,5]. The main bonding phases of WC-based cemented carbide include Co, Ni, Fe and high-entropy alloys [6,7]. Especially, Ni as a binder for WC has the advantages of good corrosion resistance and high oxidation resistance, and it has been widely studied by researchers [8,9].

The mixing uniformity of WC and the binder plays an important role in the performance of cemented carbide. Therefore, many scholars have carried out a lot of research on powder mixing methods, including electroless plating and one step co-precipitation hydrogen reduction. Electroless plating can uniformly coat a material on the surface of the substrate and is not easily affected by the surface morphology of the substrate [10]. Guo et al. [11] successfully achieving the mixing of the hard phase and the binding phase at the microscopic level. However, the activation of PdCl_2_ costs more and causes environmental pollution, Therefore, some researchers began to try self-activation pre-treatment [12] and successfully induced a subsequent electroless plating reaction by using nano Co activation sites. In the one-step co-precipitation reduction method, mixed and uniform WC binder composite powder can be obtained under the condition of reducing the grain size of WC, which is expected to be widely used in the production of cemented carbide with higher grain size requirements. Sun Yexi et al. [13] found that Co at the nanometer scale could be obtained by co-precipitation, which maintains the face-centered cubic structure and can effectively improve the toughness. Hurieh et al. [14] had also successfully prepared WC-Ni cemented carbide by co-precipitation.

Based on previous research results [12,15,16] and relevant literature investigations, it is found that electroless plating and co-precipitation methods can improve the properties of WC cemented carbide to a certain extent, but there are some differences in the effects of the two methods. In other words, there is a lack of systematic comparison of the effects of two coating methods regarding the mechanical properties and corrosion resistance under the same experimental conditions, including different binding phases of Ni and Ni/Co.

In this study, the raw material of WC-10%Ni/Co cemented carbides were mixed by using two chemical coating methods, and an appropriate amount of Co was introduced to improve the Ni agglomeration. Therefore, the effects of different chemical coating methods for the microstructure, mechanical properties and corrosion resistance were studied. The mechanism of introducing a small amount of Co to the structure of the cemented carbides with Ni as the main bonding phase was further studied.

## 2. Experimental Procedure

### 2.1. Chemicals and Raw Powders

All the specifications and manufacturers of the chemicals in this study are shown in Table 1 below.

### 2.2. Preparation of WC-Ni/Co Cemented Carbide

Figure 1 shows the schematic process for the pretreatment and plating of WC of different chemical coating methods, as described in brief as follows.

#### 2.2.1. Preparation of WC-Ni/Co^EP^ Composite Powder

The weighted WC powder was added to the coarsening solution mixed with HF (30 mL/L) and HNO_3_ (30 mL/L) and stirred for 30 min. Then, it was stood for 10 min, washed and filtered with deionized water and dried to obtain the coarse powder, which was then put into the activation solution composed with NaH_2_PO_2_·H_2_O (50 g/L) and CoCl_2_·6H_2_O (50 g/L). After that, the filtered powder was put into the Muffle furnace for 1h at 220 °C, then washed and dried to obtain the activated powder. The chemical plating solution consisted of certain amounts of CoCl_2_·6H_2_O, NiCl_2_·6H_2_O, C_6_H_5_Na_3_O_7_·2H_2_O, NaH_2_PO_2_·H_2_O and H_3_BO_3_. After adding the activated WC powder to the solution, the appearance and disappearance of obvious bubbles marked the beginning and end of the reaction, respectively. Finally, the powder was obtained after filtered, washed and dried.

#### 2.2.2. Preparation of WC-Ni/Co^CP^ Composite Powder

The coarse WC powder was added to the solution of Ni^2+^/Co^2+^ metal ions mixed with NiCl_2_·6H_2_O (0.372 mol/L) and CoCl_2_·6H_2_O (0.372mol/L), the water bath temperature of which was 50 °C. The precipitant of (NH_4_)_2_C_2_O_4_·H_2_O (0.4 mol/L) was obtained by heating and stirring it in a water bath, and then we added it to the solution containing WC powder at 1800s, and the reaction lasted for 30 min. After the stirring, we allows it to stand for 10 min, and the precipitated mixed solution was pumped and filtered, and then dried in the vacuum drying oven to obtain the mixture precursor. Additionally, the powder was obtained after it was held for 30 min at 400 °C in hydrogen protected atmosphere.

The mass ratios of WC to Ni to Co in different composite powders are shown in Table 2.

#### 2.2.3. Sintering of Composite Powder

The mixed powder of 0.5wt.% polyvinyl alcohol aqueous (5 wt.%) and 99.5 wt.% WC-Ni/Co composite powder was placed in an oven at 60 °C to dry for 1 h, and then screened through a 200 mesh sieve. A uniaxial powder tablet press was used to press and form the mixed raw materials at room temperature under a pressure of about 150 MPa. The time required for the pressure holding process was 5 min. After pressing, the long sample with the embryo body of 6.5 × 8 × 25 (mm) was put into the vacuum sintering furnace for sintering at 1375 °C for 60min.

### 2.3. Microstructural Characterization

The phase composition of the powders and the compacts were characterized by X-ray diffraction (XRD, BrukerAXS-D8 from Germany Brock AXS Co. LTD. and X-ray wavelength with Cu-Kα radiation) at a rate of 10°/min from 5 to 90° at a voltage of 40 kV, a current of 40 mA and a test accuracy of ≤0.02°, and then analyzed using MDI-JADE6.5 software. The microstructural morphology and the elemental distribution were observed by optical microscopy (OM, OLYMPUS GX41, Tokyo, Japan), scanning electron microscopy (SEM, Hitachi SU8020, Hitachi in Tokyo, Japan) equipped with energy dispersive spectroscopy (EDS) and transmission electron microscopy (TEM, FEI Tecnai G2 F30, Hillsboro, OR, USA).

The Archimedes drainage method was selected, as shown in Equation (1), to test the actual density of cemented carbide, and the instrument used was AR124CN electronic balance manufactured by Auhaus Instrument (Changzhou, China) Co., Ltd.
(1)$$\rho =\frac{m_{1}{\;\times \;\rho }_{w}}{m_{1}\;{-\;m}_{2}}$$

*ρ* is the actual density of cemented carbide (g/cm^3^);*m*_1_ is the mass of cemented carbide in air (g);*m*_2_ is the mass of cemented carbide immersed in water (g).*ρ_W_* is the density of the liquid used in the test. Pure water (1 g/cm^3^) was used in this experiment.

The relative density is the ratio of the actual density to the theoretical density of the cemented carbide sample.

### 2.4. Mechanical Performance Characterization

In this study, 200HRS-150 Rockwell hardness tester manufactured by China Shaanxi Xi’an Huayin Co., Ltd., Xi’an, China was used to measure the hardness value (HRA) of the cemented carbide. Three tests were averaged for each average value.

The impact toughness was tested using a JB-300 manual pendulum instrument of Jinan Fangyuan Company, Shandong, China. The average value of impact toughness of five samples was taken as the final result. The specific experimental process was as follows: (1) in accordance with the requirements of national standard GB229-2007 “Charpy pendulum impact test Method for metal materials”, the sample was processed into a 75 mm × 8 mm × 8 mm long strip sample, and a crack with a section of 8 mm × 1 mm and a depth of 2 mm was created at the port; (2) the sample was deep cleaned and placed on the support of the test instrument after drying; (3) then, we pulled the pendulum to a certain height and released it. After the cemented carbide sample was broken, we recorded the impact energy AKU corresponding to the height raised by the pendulum.

### 2.5. Corrosion Resistance

The CS2350 electrochemical workstation produced by China’s Hubei Province Wuhan Coster Instrument Co., LTD., Hubei, China was selected to test the cemented carbide samples in the laboratory. A calomel electrode was used as a reference electrode, and a platinum electrode was an auxiliary electrode. The dynamic potential polarization curve and impedance spectrum of the samples were measured using a three-electrode system. In the test, 3.5 wt % NaCl corrosion solution was used to simulate the seawater corrosion environment. The initial potential of the polarization process was −0.8 V, the termination potential was 2.0 V and the scanning rate was 3 mV/s.

## 3. Results

### 3.1. Sintering Behavior of WC-Ni/Co Composite Powder

The composite powder with the mass ratio of Ni/Co (mole content estimate) of 3/1 and 10% of the total mass of the cemented carbide were prepared by electroless plating (EP) and co-precipitation-high temperature reduction (CP). The morphology of the raw WC powder before coating is shown in Figure 2a. The original WC powder with a particle size of approximately 4.8~5.2 μm exhibits a smooth surface and a near-spherical grain with an irregular shape. After the Ni/Co electroless plating reaction, the surface of the WC powder was coated with a continuous layer of small balls, whose mean grain diameter is 0.266 μm, forming a rough coating “film” layer, as shown in Figure 2b. Figure 2c shows the WC-Ni/Co composite powder (WC-Ni/Co^CP^) obtained by co-precipitation-high temperature reduction. It can be clearly observed that the film layer of the bonding phase continuously and smoothly covers the surface of WC particles. In addition, the Ni/Co phase distribution is highly even, and a high-density Ni/Co phase was recorded on the surface of the WC particles. As shown in Figure 2, the binder phase particles prefer to disperse on the surface of the WC particles, instead of agglomerating between the WC particles. Meanwhile, in order to study the effect of proper Co addition on the properties of the WC-Ni cemented carbide, WC-10Ni (mass ratio) composite powder was prepared by the chemical coating method as a contrast.

Figure 3 shows the XRD of the WC-Ni/Co cemented carbide sintered after mixing it with two chemical coating methods. In addition to the main WC peaks at (001), (100), (101), (110), (002), (111), (200) and (102), a new small diffraction peak appears near 45°(the green dotted line). According to the analysis using the JADE6.5 software, this peak is a separate characteristic peak located near 44.2° of Co (111) and 44.6° of Ni (111), indicating that Co and Ni on the WC surface had formed a single phase, and it was likely to be an Ni-Co alloy.

The actual density of the sample was tested by the Archimedes drainage method, and the relative density of the different carbide was obtained. As can be seen from Table 3, the relative density of the cemented carbide prepared by co-precipitation was relatively large, close to 100%. The relative density of the sintered WC-Ni^EP^ was the lowest, at 97.4%. When a small amount of Co was added into the bonding phase, the relative density of the cemented carbide increased to 98.6%.

The reason for this difference was that during the preparation of WC-Ni^EP^, due to the agglomeration of the Ni coating, the liquid phase flow caused by sintering was impeded, and the air inside the embryo body was not easily discharged, thus reducing the relative density. However, due to the more meticulous and uniform coating on the WC surface when a small amount of Co replaced Ni in the plating solution, the cemented carbide had fewer pores during the powder mixing, and also, had a large relative density. In addition, due to the high temperature hydrogen reduction, the bond phase was well mixed with the WC particles when co-precipitation was used to prepare the WC cemented carbide powder. That is the reason why the overall relative density of cemented carbide prepared by co-precipitation is relatively high [17].

### 3.2. Comparative Analysis of Microstructure

#### 3.2.1. Grain Size Distribution

The prepared cemented carbide bars were polished, corroded for a certain period of time, and then placed under a metallographic microscope for observation. The average grain size of the cemented carbides was shown in Table 3. It can be found that the average grain size was slightly increased compared with that of the cemented carbide with pure Ni as the bonding phase after the introducing a small amount of Co, which may be because that the introduction of Co improves the uniform coating of the bonding phase on the surface of WC. Meanwhile, due to the better wettability of Co to WC, it promotes the dissolution of WC in the bonding phase during the sintering process. In addition, the Co skeleton formed in the early phase of liquid phase sintering also promotes the migration and rearrangement process of the WC powder, and finally improves the growth of the WC grains [18]. In addition, the average grain size of WC-Ni^CP^ was smaller than that of WC-Ni^EP^.

#### 3.2.2. The Fracture Morphology

During the use of the carbide, cracks will occur when the impact load exceeds its bearable range, and the continuous expansion of the cracks will lead to the fracture and failure of cemented carbide materials [19]. Cracks are often produced from the weakest place of materials. In the cemented carbide, the region between the WC grains is the weakest, followed by the WC grains, then the WC–bond interface, and finally the bond phase [20]. Therefore, the microstructure fracture modes of cemented carbide can be divided into the following four types: intergranular fractures along the WC/WC grain (C/C), transgranular fractures along the WC grain (C), inner surface fractures between WC and the bonding phase (C/B) and ductile fractures along the bonding phase (B) [21].

Figure 4a shows the fracture morphology of the cemented carbide prepared by electroless plating. As can be seen from Figure 4(a1), most of the fracture surfaces are relatively smooth, indicating that intergranular fractures (C/C) are the main fractures of cemented carbide with Ni as the bonding phase. When the bonding phase is composed of Ni/Co, the rough grain surface (C) and grain surface with ductile grain (C/B) increase, and a small dimple pattern (B) appears, the internal pores decrease also, indicating that the cemented carbide with Ni/Co as the bonding phase has a stronger resistance to impact load, and it is also more tough.

The fracture morphology of cemented carbide with different bonding phases prepared by co-precipitation hydrogen reduction is shown in Figure 4b. The characteristics of different fracture types with the bonding phase are basically the same as those of the cemented carbide prepared by electroless plating. However, it can be found that the cemented carbide obtained by co-precipitation-high temperature reduction has a higher overall density and more obvious porosity (P) reduction. In addition, abnormal grain growth can be observed in the cemented carbide with Ni as the bonding phase.

#### 3.2.3. Binder Phase Structure

Figure 5 shows the TEM morphology of WC-Ni/Co cemented carbide. The TEM image of WC-Ni/Co^EP^ is shown in Figure 5a, while the image of WC-Ni/Co^CP^ is shown in Figure 5b. It can be seen from the boundary that the diffraction at the interface of the sintered cemented carbide presents obvious transition characteristics. The sintered WC exists in a dense, hexagonal form (Figure 5(a1)), and the bonding phase has two forms, one is a densely packed, hexagonal deposited Ni-Co phase (Figure 5(a3)), the other is a face-centered, cubic deposited Ni-Co phase (Figure 5(a4)). This may be due to the large amount of WC dissolved in the Ni/Co phase, resulting in a large number of dislocations and subcrystalline structures in the cemented carbide bonding phase. Face-centered cubic (FCC) Co can stably exist in nano-sized cemented carbide without transforming into an HCP structure. Compared with HCP-Co, FCC-Co has more slip systems and is more prone to plastic deformation [22]. Cemented carbide with an FCC structure has better strength and toughness properties [23]. A further observation is that there are a few irregularly distributed diffraction spots in the bonding phase of face-centered cubic accumulation. The EDS analysis of the constituency at point e shows that there is a small amount of P element (Figure 5(a5)). Meanwhile, it can be observed that the crystal structure of the material on both sides of the interface is obvious. The Ni-Co alloy in the bonding phase also has dense hexagonal stacking and face-centered cubic stacking of WC-Ni/Co^CP^, as shown in Figure 5(b3,b4), and there are no other diffraction spots.

### 3.3. Mechanical Properties

The mechanical properties of different cemented carbides are shown in Table 3. The Rockwell hardness counter was used to measure the hardness values of the samples. The hardness of cemented carbide with Ni as the bonding phase is the lowest. Among them, the hardness of WC-Ni^CP^ is only 82.9 HRA. When we added a small amount of Co to the Ni bonding phase, the hardness greatly improved, which reached 84.3 HRA and was higher than that of the previous work with WC-18.8Ni of 82.6HRA [15]. The overall hardness of WC-Ni/Co^EP^ is higher than that of WC-Ni/Co^CP^, and the variation trends with the types of bonding phases are basically similar. This difference may be caused by the interaction of WC and the bonding phase and the introduction of P by the side reaction of electroless plating during sintering.

The flexural strength of cemented carbide with different bonding phases are also shown in Table 3. It can be seen that the flexural strength of cemented carbide with Ni as the bonding phase is the lowest, which is less than 1000 MPa, and the flexural strength of WC-Ni^CP^ is only 753 MPa. When Co was added into the bonding phase, the flexural strength of the WC-Ni/Co^EP^ improved to 938 MPa, while the flexural strength of WC-Ni/Co^CP^ improved to 1108 MPa. It shows that the introduction of a small amount of Co can enhance the flexural strength of WC cemented carbide with Ni as the main bonding phase. This phenomenon is more obvious in the process of co-precipitation hydrogen reduction, and the flexural strength increases by 47%. The bending strength of the brittle material of cemented carbide is related to the content of critical dimension defects. When Ni is used as the bonding phase, the smaller sintered WC grain size is obtained, thus the inner boundary area of cemented carbide is larger under the same volume. At this time, the grain size becomes the main factor affecting the bending strength of cemented carbide. That is the reason why the TRS of WC-Ni in EP is bigger than it is in CP. After the introduction of Co, the grain size of WC prepared by two chemical methods increases. In this case, the influence of grain size on the bending strength of cemented carbide decreases. At the same time, the high-temperature reduction of CP is very conducive to the plastic flow and uniform distribution of two-phase materials in the sintering process. In contrast, the preparation of the mixed powder by EP has higher requirements to the bonding phase powder, and the uniformity of bond phase dispersion becomes the main factor affecting the bending strength of cemented carbide at this time. Therefore, the TRS of WC-Ni/Co is bigger in CP than it is in EP [10].

Impact toughness represents the ability of the cemented carbide to absorb plastic deformation and resist fracture under impact loads, which is closely related to the fine defects in the material. In this study, the pendulum test was used to calculate the impact toughness value of cemented carbide by different chemical coating methods. The detailed experimental results are shown in Table 3. It can be seen that the impact properties of cemented carbide with Ni as the bonding phase are poor. After the addition of a small amount of Co, on the one hand, due to the good wettability of Co for WC, the two-phase material has a good plastic flow in the sintering process, which contributes to the uniform distribution of WC and the bonding phase. On the other hand, when a small amount of Co is introduced into the bonding phase, it can be strengthened by a solid solution with Ni, which can strengthen the mechanical properties of the bonding phase. Therefore, the impact toughness of cemented carbides has been greatly improved, and the impact toughness of WC-Ni/Co^CP^ has been significantly improved compared to those of WC-Ni/Co^EP^, reaching 33 KJ/m^2^, which is because that the particle size of the metal prepared by hydrogen reduction is small, at the nanometer level, which helps to retain the more face-centered cubic structure [13], so it can significantly improve the toughness of the alloy.

In summary, the improvement phenomenon in the process of preparing composite powder by co-precipitation-high temperature reduction was more significant. Due to the existence of Ni and Co in the form of Ni-Co alloy after sintering, it had a solid solution-strengthening effect, leading to improvements of the strength from 753 MPa to 1110 MPa, the toughness from 24 kJ/m^2^ to 33 kJ/m^2^ and the hardness from 82.9 HRA to 84.3 HRA.

The Table 4 lists the data comparison of various mechanical properties between this work and WC-based cemented carbides prepared by predecessors.

### 3.4. Corrosion Resistance

#### 3.4.1. Potentiodynamic Polarization

Figure 6 shows the potentiodynamic polarization curve (Tafel) of the cemented carbides with Ni and Ni/Co bonding phases. It can be seen that the current density (I) of the cemented carbide firstly decreases with the increase in the potential (V) and reaches the minimum value at around −0.2 V, and then begins to increase with the increase in the potential. It is worth noting that when the potential goes beyond −0.2 V, the cemented carbide prepared by electroless plating presents a “steep slope” (passivation phenomenon) earlier in the corrosion process. In this stage, the corrosion currents of WC-Ni^EP^ and WC-Ni/Co^EP^ change only a little with the corrosion potential, and the overall corrosion current remains at a relatively low level, which indicates that the cemented carbide prepared by electroless plating has good corrosion resistance.

Self-corrosion potential (E_corr_) represents the corrosion tendency of materials, and the smaller the value is, the easier it is to be corroded. The self-corrosion current (I_corr_) represents the corrosion rate of materials, and the higher the value is, the easier it is to be corroded. Therefore, the two parameters are important to characterize the corrosion performance of materials. Table 5 shows the electrochemical parameters of the cemented carbide under simulated seawater corrosion. It can be seen from the table that the self-corrosion potential of WC-Ni^EP^ and WC-Ni/Co^EP^ varies a little between −0.21 V and −0.25 V, and the self-corrosion current fluctuates from 7 × 10^−7^ to 8 × 10^−7^ A·cm^−2^. In other words, the self-corrosion potential value of WC-Ni^EP^ and WC-Ni/Co^EP^ is lower, indicating that it has a greater corrosion tendency. However, due to its low self-corrosion current, the corrosion rate of it is lower in the corrosion process, and the corrosion resistance is better.

#### 3.4.2. The Impedance Spectrum

In order to study the corrosion resistance of WC-Ni/Co cemented carbides, electrochemical tests were carried out. Figure 7 shows the impedance spectrum of cemented carbide in simulated seawater (Nyquist). Studies have shown that the radius of the impedance spectrum curve is positively correlated with the resistance of the tested sample, and the larger the radius is, the larger the resistance value is, and the better the corrosion resistance of the material is [25]. It can be seen from Figure 7 that the impedance radius of the WC-Ni^EP^ and WC-Ni/Co^EP^ is significantly larger than that of WC-Ni^CP^ and WC-Ni/Co^CP^, indicating that the cemented carbide prepared by electroless plating has better corrosion resistance, which is in good agreement with the polarization test results. By comparing the impedance of two cemented carbides with different bonding phases, the relationship of impedance radius is as follows: WC-Ni > WC-Ni/Co, that is, WC-Ni cemented carbides have better corrosion resistance.

Figure 8 shows the equivalent circuit diagram of the electrochemical process simulation. R_s_ represents the solution resistance determined by the corrosive medium solution of the electrochemical test, and R_p_ represents the sample resistance. The fitting circuit parameters of cemented carbide prepared by different chemical coating methods are shown in Table 2. It can be seen that the resistance of cemented carbide prepared by electroless plating is significantly greater than that prepared by co-precipitation hydrogen reduction, which is in good agreement with the conclusion obtained during the previous metallographic testing. In addition, the maximum resistance of the samples is 1.26 × 10^5^ ω of WC-Ni^EP^. However, the resistance of the cemented carbide decreased after we added a small amount of Co in Ni.

#### 3.4.3. Corrosion Resistance Mechanism

In the seawater, due to the potential difference between the metallic phase (Ni/Co) and the hard phase (WC) in the cemented carbide, the metal at the anode will lose electrons and undergo an oxidation reaction, as shown in Equations (2) and (3). This will cause the metal to dissolve and fall off from the base material:Ni → Ni^2+^ + 2e^−^(2)
Co → Co^2+^ + 2e^−^(3)

A reduction reaction occurs at the cathode, and the hard phase interacts with water and air in the environment. The specific reaction formula is as follows:O_2_ + 2H_2_O + 4e^−^ → 4O^−^(4)
W + 8OH^−^→ 4H_2_O + WO_4_^2−^+6e^−^(5)

Combined with the Tafel curves and Nyquist spectra of the cemented carbides of different specifications in 3.5wt % seawater, it can be seen that the cemented carbide with Ni as the bonding phase have excellent corrosion resistance, while that of Co has poor corrosion resistance. This is because in a neutral medium, it is easier for Co to oxidize. Previous research shows that the solid solution of metal elements in the bond phase plays an important role in enhancing the corrosion resistance of the cemented carbide. The higher the content of solid soluble elements in the bond phase is, the stronger the corrosion resistance of the cemented carbide is [26]. In the sintering process of cemented carbides with Ni as bonding phase, C atoms and W atoms in the WC matrix are more likely to diffuse into the Ni bonding phase to form a solid solution, which may also be one of the reasons for the better corrosion resistance of the WC-Ni cemented carbide [27]. The excellent corrosion resistance of cemented carbide prepared by electroless plating has a great relationship with the introduction of P in the powder coating process. It can be found in many studies that Ni-P compounds have excellent oxidation resistance and wear resistance [28]. In the process of preparing composite powder by electroless plating, the self-decomposition of reducing agent leads to a small amount of residual P. After high temperature sintering, the Ni-Co-P alloy is formed by the interaction of the P element with Ni and Co, which can prevent the oxidation dissolution of the bonding phase in the corrosion process, thus improving the corrosion resistance of cemented carbide.

## 4. Conclusions

WC-10% (Ni and Ni/Co) composite powders were, respectively, prepared by electroless plating and co-precipitation, and then consolidated for a comparison of the microstructures, mechanical properties and corrosion resistance. The conclusions are obtained as follows.

(1)The Ni/Co bonding phase can be continuously and compactly coated on the surface of WC particles after mixing the powder by electroless plating and co-precipitation hydrogen reduction methods.(2)The density and grain size of CP were denser and finer than those of EP after sintering. Due to the solid solution strengthening effect of Ni-Co alloy, the bending strength and impact toughness of WC-Ni/Co^CP^ were improved to 1110 MPa and 33kJ/m^2^, respectively.(3)The simulated resistance value of WC-Ni^EP^ can be maintained at about 1 × 10^5^ Ω due to the presence of the Ni-Co-P alloy, while the corrosion resistance of which was also increased up to 9.018 × 10^4^ Ω, indicating the excellent corrosion resistance.

## Figures and Tables

**Figure 1 materials-16-01977-f001:**
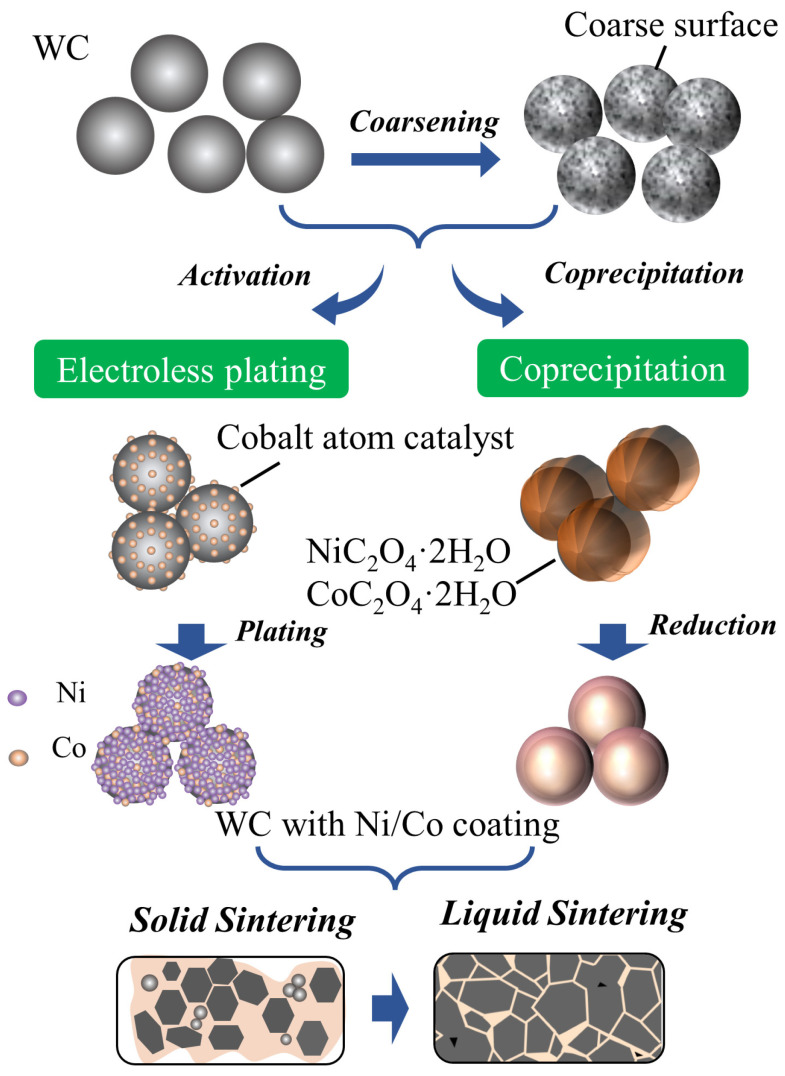
Preparation of WC-Ni/Co composite powder and densification.

**Figure 2 materials-16-01977-f002:**
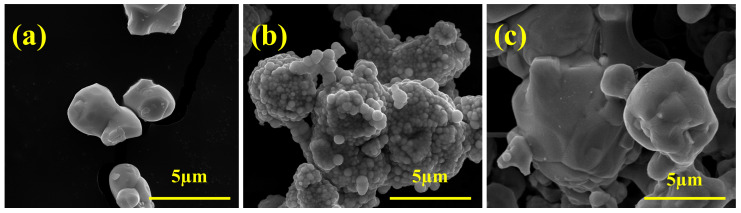
SEM morphology of WC-Ni/Co cemented carbide: (**a**) raw WC; (**b**) WC-Ni/Co^EP^; (**c**) WC-Ni/Co^CP^.

**Figure 3 materials-16-01977-f003:**
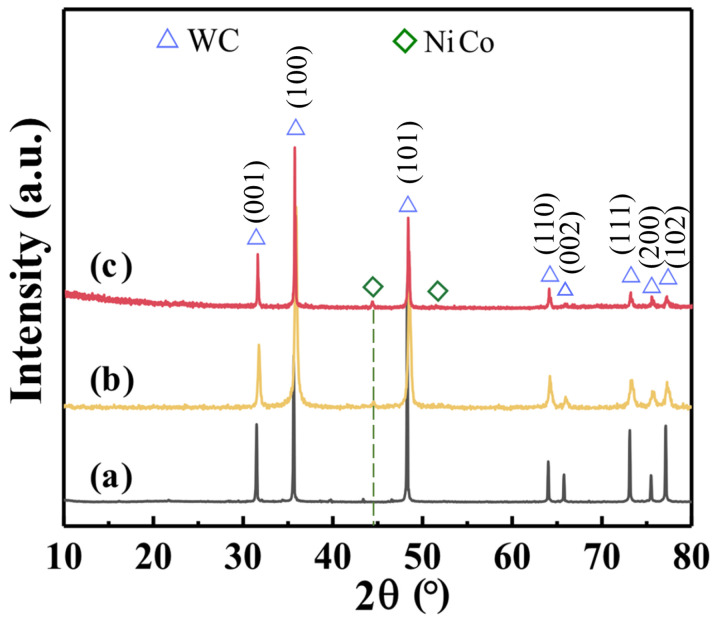
The XRD of sintered cemented carbide: (a) raw WC; (b) WC-Ni/Co^EP^; (c) WC-Ni/Co^CP^.

**Figure 4 materials-16-01977-f004:**
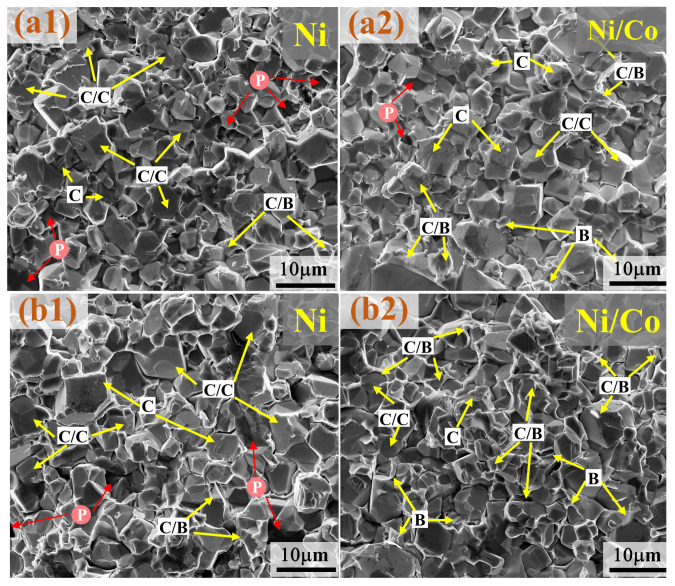
Fracture morphology of WC-Ni/Co cemented carbide: (**a1**,**a2**) WC-Ni/Co^EP^; (**b1**,**b2**) WC-Ni/Co^CP^.

**Figure 5 materials-16-01977-f005:**
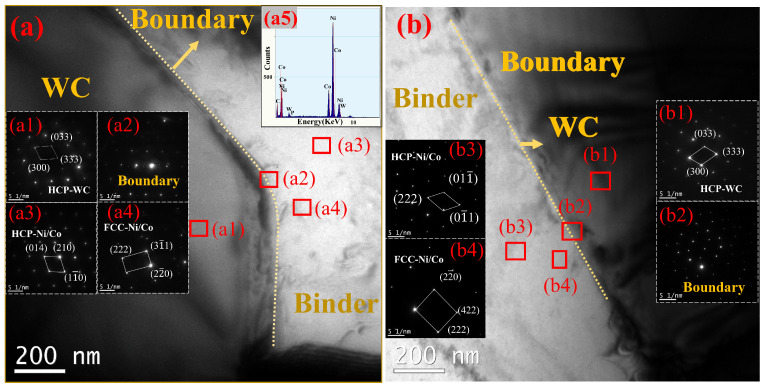
The TEM image of WC-Ni/Co cemented carbide: (**a**) WC-Ni/Co^EP^; (**b**) WC-Ni/Co^CP^, in which (**a1**) is black zone diffraction; (**a2**) is two phase interface diffraction; (**a3**,**a4**) are the white area diffraction; (**a5**) is the eds in a4; and also (**b1**) is black zone diffraction; (**b2**) is two phase interface diffraction; (**b3**,**b4**) are the white area diffraction.

**Figure 6 materials-16-01977-f006:**
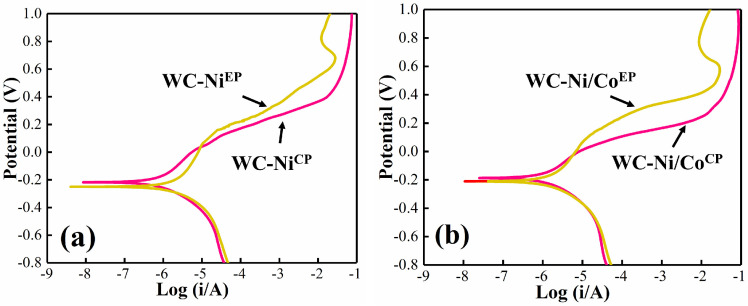
Polarization curves of cemented carbide samples: (**a**) WC-Ni^EP^ and WC-Ni^CP^; (**b**) WC-Ni/Co^EP^ and WC-Ni/Co^CP^.

**Figure 7 materials-16-01977-f007:**
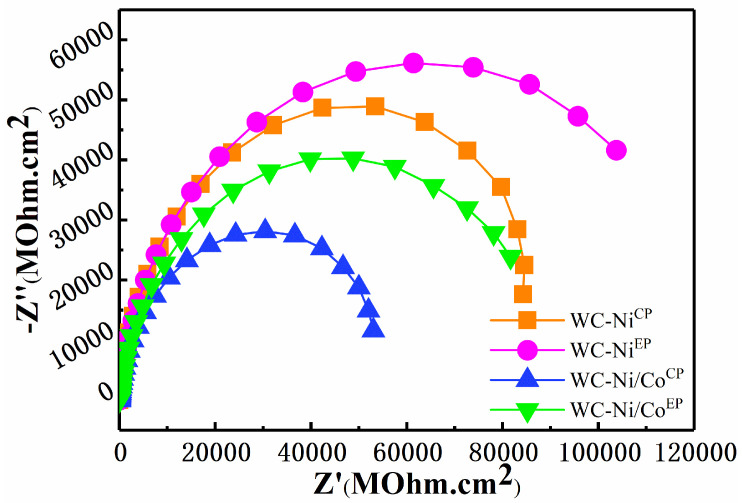
Impedance spectra of cemented carbide samples.

**Figure 8 materials-16-01977-f008:**
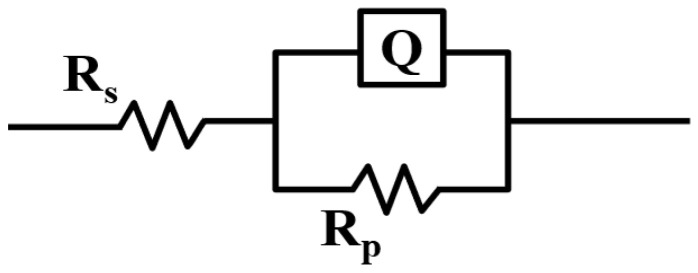
Impedance test EIS equivalent circuit diagram (R_S_: solution resistance; R_p_: sample resistance).

**Table 1 materials-16-01977-t001:** Raw material parameter table.

Serial Number	Material Name	Material Specification	Manufacturer
1	WC substrate powder	4.8~5.2 μm	Sichuan Zigong Co., Ltd.
2	HF	500 mL	Sinopharm Chemical Reagent Co., Ltd.
3	HNO_3_	500 mL	Sinopharm Chemical Reagent Co., Ltd.
4	NaH_2_PO_2_·H_2_O	AR 500 g	Sinopharm Chemical Reagent Co., Ltd.
5	C_6_H_5_Na_3_O_7_·2H_2_O	AR 500 g	Sinopharm Chemical Reagent Co., Ltd.
6	H_3_BO_3_	AR 500 g	Sinopharm Chemical Reagent Co., Ltd.
7	NaOH	AR 500 g	Sinopharm Chemical Reagent Co., Ltd.
8	HCl	AR 500 g	Sinopharm Chemical Reagent Co., Ltd.
9	NiCl_2_·6H_2_O	AR 500 g	Shanghai Lingfeng Co., Ltd.
10	CoCl_2_·6H_2_O	AR 500 g	Shanghai Lingfeng Co., Ltd.
11	(NH_4_)_2_C_2_O_4_·H_2_O	AR 500 g	Shanghai Lingfeng Co., Ltd.

**Table 2 materials-16-01977-t002:** Different element mass fraction ratios in composite powders.

Powder and Element	WC-Ni^EP^	WC-Ni^CP^	WC-Ni/Co^EP^	WC-Ni/Co^CP^
WC (wt.%)	90	90	90	90
Ni (wt.%)	10	10	7.5	7.5
Co (wt.%)	/	/	2.5	2.5

**Table 3 materials-16-01977-t003:** Sintering and mechanical properties of cemented carbide.

Sintering and Mechanical Properties	WC-Ni^EP^	WC-Ni^CP^	WC-Ni/Co^EP^	WC-Ni/Co^CP^
Relative density (%)	97.41	99.80	98.59	99.87
Average grainsize(µm)	4.77	4.13	5.23	4.59
Hardness (HRA)	84.4	82.9	86.5	84.3
Flexural strength (MPa)	857	753	938	1108
Impact toughness (KJ/m^2^)	12	24	20	33

**Table 4 materials-16-01977-t004:** Comparison of mechanical properties of traditional WC-based cemented carbide.

Material	WC Grain Size (µm)	Hardness	TRS (MPa)	Impact Toughness	Ref.
WC-12Co	2–17	81 HRA	650	/	[13]
WC-12Co	50 (Maximum)	>1750 HV	600–800	/	[24]
WC-10Co	7.3	84.2 HRA	1752	34.5	[17]
WC-18.8Ni	/	82.6 HRA	/	/	[15]
WC-10(Ni, Co)	4.59	84.3 HRA	1108	33	This work

**Table 5 materials-16-01977-t005:** Electrochemical and fitting circuit parameters of cemented carbide in simulated seawater.

Cemented Carbide	I_corr_/(A·cm^−2^)	E_corr_/(V)	R_s_/(Ω)	R_p_/(Ω)
WC-Ni^EP^	8.1686 × 10^−7^	−0.25018	5.739	1.26 × 10^5^
WC- Ni^CP^	3.6848 × 10^−6^	−0.21734	9.027	9.184 × 10^4^
WC-Ni/Co^EP^	6.9858 × 10^−7^	−0.21039	5.268	9.018 × 10^4^
WC-Ni/ Co^CP^	8.8971 × 10^−6^	−0.18726	6.674	5.25 × 10^4^

## Data Availability

The data that support the findings of this study are available from the corresponding author: Jianfeng Zhang, upon reasonable request.

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
