# Peer review of "Comparative Study on the Densification, Microstructure and Properties of WC-10(Ni, Ni/Co) Cemented Carbides Using Electroless Plated and Coprecipitated Powders"

_materials, 2023, doi:10.3390/ma16051977_

Round 1

Reviewer 1 Report

The work is of technical interest. However, the issues are still noticed.

1. English is not fine.

2. Title should be changed to a precise one.

3. Introduction part is not sufficient.

3. Authors preferred conventional sintering at high temperatures. But microwave sintering requires small temperatures. Justify your answer.

4. How to measure the densities for the present samples?

5. What is the reason behind cluster like morphology?

6. Is there any electron exchange between two cations of current samples?

Reviewer 2 Report

Manuscript ID: materials-2191052

Comparative study on the densification, microstructure and properties of WC-Ni/ Co cemented carbides using electroless plated and coprecipitated powders

The manuscript entitled "Comparative study on the densification, microstructure and properties of WC-Ni/ Co cemented carbides using electroless plated and coprecipitated powders" was reviewed. This work and as-obtained results are interesting. In this study, WC-10%Ni/Co cemented carbides raw material were mixed by two chemical coating methods to improve the Ni agglomeration in traditional ball milling process, and appropriate amount of Co was introduced to strengthen the cemented carbide in the process of coating. Therefore, the influence of different chemical coating methods and binding phase on the mechanical properties and corrosion resistance of the cemented carbide was studied. The mechanism of introducing a small amount of Co on the structure of the cemented carbides with Ni as the main bonding phase was further studied. I have the following comments;

1. Abstract should have some numerical data. The Abstract part is weak and it must be more informative by including more mathematical findings and more powerful explanations.

2. The idea of the research seems to be interesting but the set goals are not achieved. What the main significance of paper in comparison is of relates published works?

3. Introduction writing part is not satisfactory. Need to be improved. The whole generalization for this paper should be given in the introduction.

4. Write the novelty of the work clearly in the introduction section.

5. The authors should enhance the discussion and comparison with the results in literature.

6. I have read and evaluated the manuscript and in my opinion the submission does not yet sufficiently justify publication. Discuss the shortcomings of previous work and the gaps and how this work intends to fill those gaps. Related references should be cited:

Journal of Molecular Catalysis A: Chemical 186 (1-2) (2002) 101-107;

Journal of Molecular Catalysis A: Chemical 201 (1-2) (2003) 43-54; 

Journal of Molecular Catalysis A: Chemical 245 (1-2) (2006) 192-199

7. Every data should have a discussion and conclusion beyond simple description. The authors should add a full and comprehensive discussion to all parts as well.

8. In Experimental part Authors should list all the reagents used, including their purity, supplier, etc.

9. The peaks in XRD patterns should be indexed.

10. The EIS fitting curve is necessary to show how the fitting comes.

11. Please upload the raw data for the EIS plots, and also please describe a relative relationship between CV and EIS analysis?

12. The structure of the manuscript might need a major adjustment for a better understanding.

13. The present form was not satisfactory for publication, authors should analyze the results intensively in the aspect of XPS elementary positions using deconvolution parameters.

14. The Conclusions section is too long. It should be kept short and must be fully supported by the results reported.

I recommend publication of this article after major revisions and would like to see the revised version of paper before publication.

Author Response

Please see the attachment. We are happy to upload the original EIS data, but we regret that the reply page does not allow multiple files to be uploaded.

Reviewer 3 Report

The article has an acceptable quality. However, the following aspects should be addressed before publication:

1. I would like to see a comparison between EP, CP and planetary ball milling. I want to see if EP and CP are better mixing methods than planetary ball milling.

2. A table with the wt.% content of W, C, Ni and Co would be useful. Also, the grain size of these elements can be interesting.

3. Why is there a difference in the roughness of WC-Ni/Co in EP and CP? Is there any specific reason?

4. Why is bigger the TRS of WC-Ni in EP than in CP? However, in WC-Ni/Co the TRS is bigger in CP than EP. Can you explain this?

5. The addition of Co increases the density in CP and EP. Why?

6. The addition of Co increases the grain size. Why?

7. The grain size of WC-Ni in EP is higher than in CP. Why?

8. It is difficult to see well the fracture modes in Figure 4. Can you improve the quality of the figure? Also, is there any way to avoid the abnormal grain growth in the case of cemented carbide with Ni as bonding phase? Which is the reason of the abnormal grain growth in this case?

Reviewer 4 Report

1. Literature studies on the subject of the manuscript are presented very limited. In addition, the novelty of the study from other studies in the literature have not been fully presented. In this context, the introduction should be expanded.

2. Coating thicknesses are not given. For this, cross-sectional images should be taken.

3. How many times were the corrosion tests repeated? Also, why was salt water used in the tests? Which field do the authors consider as the application field?

4. The chemical properties of the purchased powder can be given in a table under the heading in line 74.

5. How was the Ni ratio in the WC-10Ni powder specified in line 170 calculated?

6. In line 118, it should be stated which tube (Cu or Co) the XRD process was used for.

7. How were the sample sizes specified in line 143 determined? Is there a standard?

8. In the XRD graph given in line 180, the peak at approximately 67° is not shown. This peak should be indicated on the graph as the WC peak.

9. Is the 620Mpa flexural strength value specified in line 384 and abstract line 21 correct? Shouldn't it be 753 MPa?

10. In Figure 6, the polarization curve for the WC-Ni (CP) sample is not visible.

Round 2

Reviewer 2 Report

ACCEPT

Round 3

Reviewer 2 Report

ACCEPT